# Implicit Transformer Network for Screen Content Image Continuous Super-Resolution

**Jingyu Yang**[1]     **Sheng Shen**[1]     **Huanjing Yue**[1*]    **Kun Li**[2]

[1]School of Electrical and Information Engineering, Tianjin University
[2]College of Intelligence and Computing, Tianjin University
{yjy, codyshen, huanjing.yue, lik}@tju.edu.cn
https://github.com/codyshen0000/ITSRN

## Abstract

Nowadays, there is an explosive growth of screen contents due to the wide application of screen sharing, remote cooperation, and online education. To match the limited terminal bandwidth, high-resolution (HR) screen contents may be downsampled and compressed. At the receiver side, the super-resolution (SR) of low-resolution (LR) screen content images (SCIs) is highly demanded by the HR display or by the users to zoom in for detail observation. However, image SR methods mostly designed for natural images do not generalize well for SCIs due to the very different image characteristics as well as the requirement of SCI browsing at arbitrary scales. To this end, we propose a novel Implicit Transformer Super-Resolution Network (ITSRN) for SCISR. For high-quality continuous SR at arbitrary ratios, pixel values at query coordinates are inferred from image features at key coordinates by the proposed implicit transformer and an implicit position encoding scheme is proposed to aggregate similar neighboring pixel values to the query one. We construct benchmark SCI1K and SCI1K-compression datasets with LR and HR SCI pairs. Extensive experiments show that the proposed ITSRN significantly outperforms several competitive continuous and discrete SR methods for both compressed and uncompressed SCIs.

## 1 Introduction

Nowadays, screen content images are becoming ubiquitous due to the wide application of screen sharing and wireless display. Meanwhile, due to the limited bandwidth, screen content images received by users may be in low-resolution (LR) and users may need to zoom in the content for detail inspection. Therefore, screen content image super-resolution (SCI SR) is to improve the quality of LR SCIs.

However, different from natural scene images, SCIs are dominated by the contents generated or rendered by computers, such as texts and graphics. Such contents highly demanded are characterized by thin and sharp edges, little color variance, and high contrast. In contrast, the natural scene images are relatively smooth, and contain rich colors and textures. Conventional image SR methods designed for nature images are good at modeling the local smoothness of natural images other than the thin and sharp edges in SCIs. Very recently, Wang *et al.* [1] proposed a SR method for compressed screen content videos, which addressed the compression artifacts of screen content videos by introducing a distortion differential guided reconstruction module. However, their network is still composed of fully convolution layers without designing specific structures for the thin and sharp edges in screen contents. In addition, they utilize previous frames to help reconstruct the current frame, which makes it unsuitable for frame-wise SCI SR.

---

[*]*Corresponding author: Huanjing Yue*

35th Conference on Neural Information Processing Systems (NeurIPS 2021).

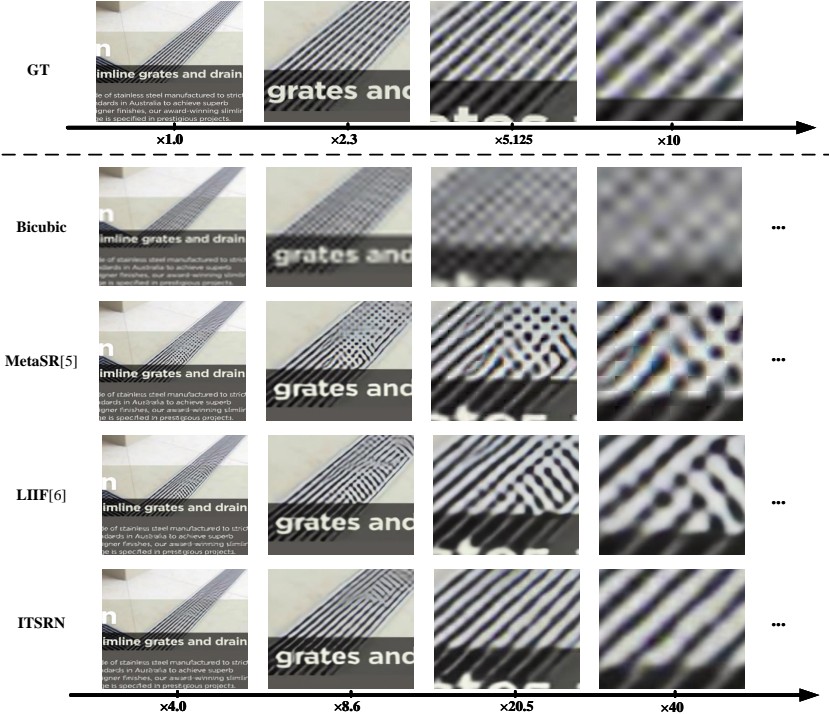

Figure 1: Comparison of the proposed Implicit Transformer SR Network (ITSRN) with state-of-the-art image continuous magnification methods. The ground truth (GT), which has the same resolution with that of the ×4 upsampling, is visualized at the top row, and its magnification results (×2.3, ×5.125, ×10) are obtained by bicubic-interpolation.

On the other hand, conventional SR methods are designed for discrete (*i.e.,* several fixed) magnification ratios [2, 3, 4], making them hard to fit screens with various sizes. Recently, a few SR methods for continuous magnification have been proposed [5, 6]. Hu *et. al.* [5] proposed to perform arbitrary-scale SR with a learnable up-sampling weight matrix based on meta-learning. The work LIIF [6] introduced the concept of implicit function [7, 8, 9] to image SR. The implicit function, which attempts to represent images with continuous coordinates and directly maps the coordinates to values, enables continuous magnification.

In this work, we observe that convolution filters could be harmful to sharp and thin edges in SCIs since the weight sharing strategy makes them tend to produce a smooth reconstruction result. Therefore, we propose to render the pixel values by a point-to-point implicit function, which adapts to image content according to image coordinates and pixel features. Fortunately, this also enables us to perform continuous magnification for SCIs. We would like to point out that even with the point-to-point implicit function, reconstructing dense edges are still quite challenging. As shown in Fig. 1, LIIF [7] cannot reconstruct the dense edges well since it directly concatenates the pixel coordinates and features together to predict the pixel values, which is not optimal since the two variables have different physical meanings. As a departure, we reformulate the interpolation process as a transformer and introduce implicit mapping to model the relationship between pixel coordinates, which are used to aggregate the pixel feature. Our main contributions are summarized as follows.

- First, we propose a novel *Implicit Transformer Super-Resolution Network* (ITSRN) for SCI SR. The LR and HR image coordinates are termed as the "key" and "query", respectively. Correspondingly, the LR image pixel features are termed as "value". In this way, we can infer pixel values by an implicit transformer, where implicit means we model the relationship between LR and HR images in terms of coordinates instead of pixel values.

- Second, instead of directly concatenating the coordinates and pixel features to predict the pixel value, we propose to predict the transform weights with query and key coordinates via nonlinear mapping, which are then used to transform the pixel features to pixel values. In

addition, we propose an implicit position encoding to aggregate similar neighboring pixel values to the central pixel.

- Third, we construct a benchmark dataset with various screen contents for SCI SR. Extensive experiments demonstrate that the proposed method outperforms the competitive continuous and discrete SR methods for both compressed and uncompressed screen content images. Fig. 1 presents an example of our SR results, which demonstrates that the proposed method is good at reconstructing thin and sharp edges for various magnification ratios.

## 2 Related Work

### 2.1 Screen Content Processing

The screen content is generally dominated by texts and graphics rendered by computers, making the pixel distribution of screen contents totally different from that of natural scenes. Therefore, many works specifically designed for screen contents are proposed, such as screen content image quality assessment [10, 11, 12, 13], screen content video (image) compression [14, 15]. However, there is still no work exploring screen content image SR. Very recently, Wang *et al.* [1] proposed screen content video SR, which reconstructed the current frame by taking advantage of the correlations between neighboring frames, making it cannot deal with image SR. In addition, its main motivation is solving the SR problem when the videos are degraded by compression other than designing specific structures for continuously recovering thin and sharp edges in screen contents. In this work, we address this issue by introducing point-to-point implicit transformation.

### 2.2 Continuous Image Super-Resolution

Image SR refers to the task of recovering HR images from LR observations. Many deep learning based methods have been proposed for super-resolving the LR image with a fixed scale [16, 3, 2, 17, 18, 4, 19, 20]. Since the screen contents are usually required to be displayed on screens with various sizes. Therefore, continuous SR is essential for screen contents. In recent years, several continuous image SR methods [5, 6] are proposed in order to achieve arbitrary resolution SR. MetaSR [5] introduces a meta-upscale module to generate continuous magnification but it has limited performance in dealing with out-of-training-scale upsampling factors. LIIF [6] reformulates the SR process as an implicit neural representation(INR) problem, which achieves promising results for both in-distribution and out-of-distribution upsampling ratios. Inspired by LIIF, we utilize the point-to-point implicit function for SCI SR.

### 2.3 Implicit Neural Representation

Implicit Neural Representation (INR) usually refers to continuous and differentiable function (*e.g.,* MLP), which can map coordinates to a certain signal. INR was widely used in 3D shape modeling [21, 22, 23, 24], volume rendering (*i.e.,* neural radiance fields(Nerf)) [9, 25], and 3D reconstruction [8, 26]. Very Recently, LIIF [6] was proposed for continuous image representation, in which networks took image coordinates and the deep features around the coordinate as inputs, and then map them to the RGB value of the corresponding position. Inspired by LIIF, we propose an implicit transformer network to achieve continuous magnification while retaining the sharp edges of SCIs well.

### 2.4 Positional Encoding (Mapping)

*Positional encoding* is critical to exploit the position order of the sequence in transformer networks [27, 28, 29]. We have to utilize positional encoding to indicate the order of the sequence since there are no other modules to model the relative or absolute position information in the sequence-to-sequence model. In the literature, sine and cosine functions are used for positional encoding in transformer [27]. Coincidentally, we find that the Fourier feature (combined with a set of sinusoids) based *positional mapping* is used in the implicit function to improve the convergence speed and generalization ability [30, 9, 31]. Specifically, a Fourier feature mapping is applied on the input coordinates to map them to a higher dimensional hypersphere before going through the coordinate-based MLP. Although the two schemes are proposed based on different motivations, they show significant effects in representing position information, which further boost the final results. Inspired

by them, we propose implicit positional encoding to model the relationship between neighboring pixel values. Here "implicit" means that we do not explicitly encode (map) the coordinates but encode the pixel values of neighboring coordinates.

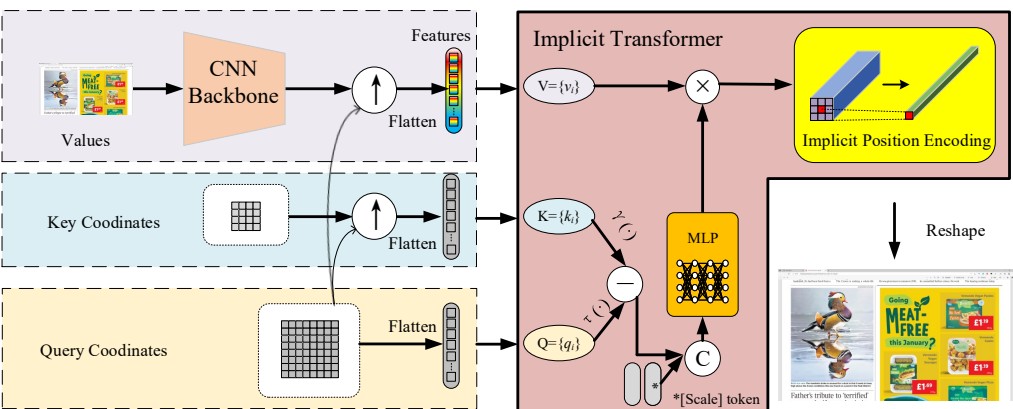

Figure 2: The framework of our proposed ITSRN. The input LR image first goes through a CNN backbone to generate pixel features. Then the features and key coordinates are upsampled by nearest neighbor interpolation based on query coordinates. Hereafter, we utilize the proposed implicit transformer to learn the transform weights with query, key coordinates and scale token, and the pixel value is obtained via transforming the pixel features with transform weights. Finally, the pixel value is further refined by the proposed implicit position encoding. The symbols $\uparrow$, $\ominus$, $\copyright$, and $\otimes$ refer to upsampling, subtraction, concatenation, and matrix product operations respectively.

## 3 Approach

Figure 2 illustrates the framework of the proposed ITSRN. In the following, we give details for the proposed implicit transformer network and implicit position encoding.

### 3.1 Implicit Transformer Network

Let's first review the process of image interpolation. Suppose we have an LR image $I^L$ that needs to be upsampled. The pixel value of query point $q(i, j)$ in the HR image $I^H$ is obtained by fusing pixel values of its closest key points $k(i', j')$ in $I^L$ with a weighting matrix. Denoted by $Q$ the query points in upsampled image, $K$ the key points in the input LR image, and $V$ the feature values on the corresponding key points. Then, the image interpolator can be reformulated as a transformer [27]. Different from the explicit transformer which takes pixel values as $Q$ and $K$, the interpolation transformer deals with pixels' coordinates instead of their values. Inspired by the implicit function in NeRF [9], which uses the pixel coordinates to generate RGB values, we reformulate the interpolation process as *Implicit Transformer*, and propose a novel Implicit Transformer Network for SCI SR.

The super-resolution process as well as image interpolation is reformulated in terms of implicit function as

$$I_q = \Phi(q, k, v), \tag{1}$$

where $I_q$ is the target RGB value that need to be predicted, $q$ is the query coordinate(s) in $I^H$, $k$ is the key coordinate(s) in $I^L$, and $v$ is the pixel value(s) or feature(s) corresponding to the key coordinate(s). Note that both $q$ and $k$ are coordinates in the continuous image domain. $\Phi$ is a mapping function which maps coordinates and features to RGB values. In image interpolation, $k$ is the neighboring key coordinate of $q$, $v$ are the pixel value of $k$, $\Phi$ is the weighting matrix. In implicit function based SR method LIIF [6], $k$ is the nearest neighbor coordinate in $I^L$ for $q$ in $I^H$, and $v$ is the corresponding pixel feature of $k$. LIIF directly concatenates $v$ and the relative coordinate of $q$ from $k$, and then utilize the nonlinear mapping $\Phi$ realized by a multi-layer perceptron (MLP) to render the pixel value $I_q$. It achieves promising results due to the strong fitting ability of the MLP. However, we observe that directly concatenating the pixel feature and the relative coordinates is not optimal since they

have different physical meanings. To solve this problem, following the idea of transformer [27], we reorganize Formula (1) as follows.

$$I_q = \Phi(q, k, v) = \phi(q, k)v. \tag{2}$$

Different from the explicit transformer, here $q$ and $k$ are pixel coordinates other than pixel values. In other words, we are not learning the the $RGB \rightarrow RGB$ mapping, but the $coordinate \rightarrow RGB$ mapping. Due to the continuity of coordinates, $\phi(\cdot)$ is a continuous projection function, which computes the transform weights $\phi(q, k)$ to aggregate the feature $v$, $i.e.$, $I_q$ is computed with the multiplication of $\phi$ and $v$. To further illustrate it, we decompose $\phi$ as:

$$\phi(q, k) = f(\delta(q, k)), \tag{3}$$

where the function $\delta$ produces a vector that represents the relationship between $q$ and $k$. Then the mapping function $f$ projects this vector into another vector, which is then multiplied with feature $v$ as indicated in Eq. 2.

There are many ways to model the relation function $\delta$, such as dot product and hadamard product commonly used in transformers [27]. Different from their approaches, in this paper, we utilize subtraction operation, $i.e.$,

$$\delta(q, k) = \gamma(q) - \tau(k), \tag{4}$$

where $\gamma$ and $\tau$ can use trainable functions or identity mapping. In this work, we utilize identity mapping since it generates similar results as that of trainable functions.

Inspired by ViT's [32] $[class]$ token, which is extra global information for classification, we augment the input coordinates with scale information, and name it as $[scale]$ token. It represents the global magnification factor. With the $[scale]$ token, the implicit transformer can take the shape of the query pixel as additional information for reconstructing the target RGB value. Although it is feasible to predict RGB values without $[scale]$ token, it is not optimal since the predicted RGB value should not be completely independent of its shape [2]. The output of $\delta(q, k)$ is concatenated with the $[scale]$ token, and is then fed to the mapping function $f$. In this way, $\phi$ in Formula (3) is reformulated as

$$\phi(q, k) = f([\delta(q, k), s]), \tag{5}$$

where $s = (s_h, s_w)$ is a two-dimensional scale token, representing the upsampling scale along the row and column for the query pixel. $[\delta(q, k), s]$ means the concatenation of $\delta(q, k)$ and $s$ along the channel dimension. Note that, to avoid large coordinates in HR space, we normalize the coordinates as

$$p(i, j) = [-1 + \frac{2i + 1}{H}, -1 + \frac{2i + 1}{W}], i \in [0, H - 1], j \in [0, W - 1], \tag{6}$$

where $(i, j)$ represents the spatial location within the $R^{H \times W}$ space.

## 3.2 Implicit Position Encoding

Although we have considered the positional relationship between $q$ and $k$, we still ignore the relative position of different query points. Therefore, inspired by the position encoding in explicit transformer, we propose to introduce implicit position encoding (IPE) to avoid the discontinuity of neighboring predictions. Here, "Implicit Position Encoding" means that we do not explicitly encode the positional relationship among the pixel coordinates in the $Q$ sequence since they are already absolute position coordinates. On the contrary, we encode the pixel value relationships within a local neighborhood. Chu $et\ al.$ claimed that convolution, which models the relationship between neighboring pixels, could be considered as an implicit position encoding scheme [33]. Therefore, we termed Eq.7 as implicit position encoding. Specifically, we unfold the query within a local window. The final predict value of the query coordinate is conditioned on its neighbor values. The IPE process is denoted as

$$\hat{I}_q = \sum_{p \in \Omega(q)} w(p, q) I_q, \tag{7}$$

where $\hat{I}_q$ is the refined pixel value, $\Omega(q)$ is a local window centered at $q$, $w$ represents the neighbors' (denoted by $p$) contribution to the target pixel. In traditional image filtering, $w(p, q)$ is generally realized by a Gaussian filter or bilateral filter. In this work, we utilize an MLP to learn adaptive weighting parameters for each $q$.

---

[2]It is surprised to find that our formulation in terms of transformer for the $[scale]$ token is similar to the cell decoding strategy in LIIF.

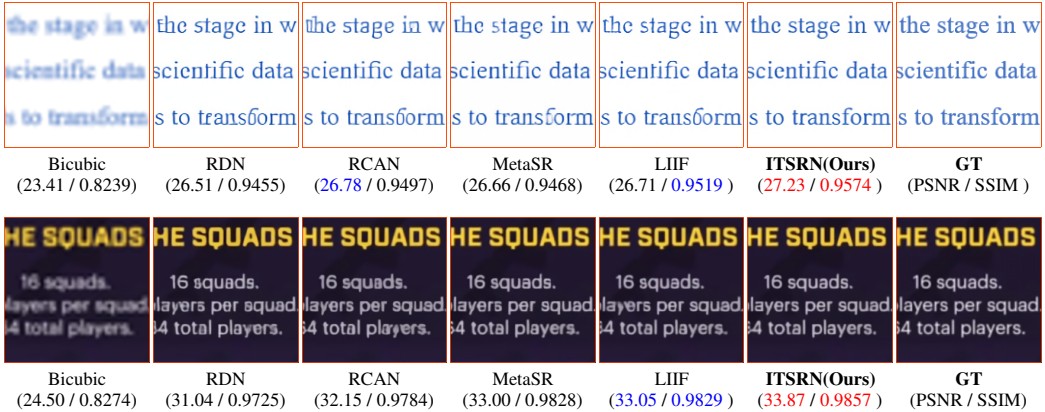

| Bicubic | RDN | RCAN | MetaSR | LIIF | ITSRN(Ours) | GT |
|---|---|---|---|---|---|---|
| (23.41 / 0.8239) | (26.51 / 0.9455) | (26.78 / 0.9497) | (26.66 / 0.9468) | (26.71 / 0.9519 ) | (27.23 / 0.9574 ) | (PSNR / SSIM ) |
| Bicubic | RDN | RCAN | MetaSR | LIIF | ITSRN(Ours) | GT |
| (24.50 / 0.8274) | (31.04 / 0.9725) | (32.15 / 0.9784) | (33.00 / 0.9828) | (33.05 / 0.9829 ) | (33.87 / 0.9857 ) | (PSNR / SSIM) |

Figure 3: Visual comparison with state-of-the-arts at ×4 SR.

## 4 Architecture Details of ITSRN

As shown in Figure 2, the proposed ITSRN has three parts, *i.e.,* a CNN backbone to extract compact feature representations for each pixel, an implicit transformer for mapping coordinates to target values, and an implicit position encoding that further enhances the target values. In the following, we give details for the three modules.

**Backbone.**    Given an LR image $I^L \in \mathcal{R}^{3 \times h \times w}$, we utilize a CNN to extract its feature map $V \in \mathcal{R}^{c \times h \times w}$. In our experiments, $c = 64$. Normally, any CNN without downsampling / upsampling can be adopted as the feature extraction backbone. To be consistent with LIIF [6], we utilize RDN [18] (excluding its up-sampling layers) as the backbone. The extracted features $V$ will be used in the following implicit transformer.

**Implicit Transformer.**    First, inspired by LIIF [6] and MetaSR [5], to enlarge each feature's local receptive field, we apply feature unfolding, namely concatenating the features for the pixels in a local region ($3 \times 3$ in this work) to get $v' \in \mathcal{R}^{c \times 9 \times 1 \times 1}$. After that, $v'$ replaces $v$ for the following processes. All the key coordinates corresponding to the feature vectors construct a coordinate matrix $K \in \mathcal{R}^{2 \times h \times w}$ with the same spatial shape as $V$. The channel dimension is 2, which includes the row and column coordinates. Similarly, the query coordinates build a matrix $Q \in \mathcal{R}^{2 \times H \times W}$, where $H \times W$ is larger than $h \times w$. Therefore, we first utilize the nearest neighbor interpolation to upsample the coordinates $K$ to the size of $Q$. Hereafter, we utilize an MLP as the nonlinear mapping function $f(\cdot)$ (mentioned in Eq. 3) to learn the relationship between the coordinate pairs in upsampled $K$ and $Q$. As demonstrated in Eq. 5, $f(\cdot)$ maps the 4-dimensional (coordinates and scale token) input to a $9c$-dimensional output($9c \times 3$ for RGB channels). Finally, the $9c$-dimensional output is multiplied by the corresponding pixel feature $v$, generating the coarse result for $I_q$.

**Implicit Position Encoding.**    After obtaining the coarse value $I_q$, we further utilize IPE to refine it. In this way, the central pixel can be more continuous with its neighbors. As mentioned in Eq. 7, we set $\Omega$ to $3 \times 3$ and $w$ for each point is learned via an MLP. The MLP includes two layers, *i.e.,* $\{Linear \rightarrow Gelu[27] \rightarrow Linear\}$, and the numbers of neurons for the two layers are 256 and 1, respectively.

## 5 Experiments

### 5.1 Datasets

To the best of our knowledge, there is still no SCI SR dataset for public usage. Therefore, we first build a dataset named SCI1K. It contains 1000 screenshots with various screen contents, including but not limited to web pages, game scenes, cartoons, slides, documents, *etc.* Among them, 800 images with 1280×720 are used for training and validation. The other 200 images with resolution ranging

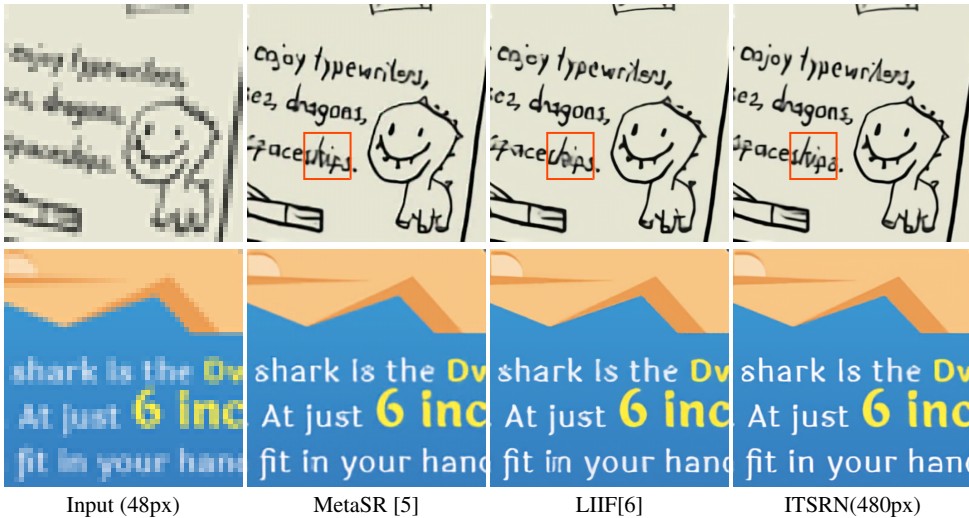

| Input (48px) | MetaSR [5] | LIIF[6] | ITSRN(480px) |

Figure 4: Visual comparison with state-of-the-arts for arbitrary SR results. The input is a $48 \times 48$ patch from an image in SCID [12] test set. All the three models are trained with continuous scales in the range $\times 1 \sim \times 4$ and are tested for $\times 10$ magnification. Note that the LR input is generated with $\times 4$ downsampling, and there is no groud truth for its $\times 10$ magnification.

from $1280 \times 720$ to $2560 \times 1440$ are used for testing. To be consistent with previous works [4, 5, 6, 18], we utilize bicubic downsampling to synthesize the LR images. In addition, to simulate the degradation introduced in transmission and storage, we build another dataset, named SCI1K-compression, by utilizing JPEG compression to further degrade the LR images. The quality factor of JPEG is randomly selected from 75, 85, and 95.

To evaluate the generalization of the trained model, besides our test set, we also test on two other screen content datasets constructed for image quality assessment, *i.e.,* SCID (including 40 images with resolution $1280 \times 720$ ) [12] and SIQAD (including 20 images with resolution around $600 \times 800$)[13].

## 5.2 Training Details

In the training phase, to simulate continuous magnification, the downsampling scale is sampled in a uniform distribution $\mathcal{U}(1,4)$. We then randomly crop $48 \times 48$ patches from the LR images and augment them via flipping and rotation.

Following [18], we utilize the $\ell 1$ distance between the reconstructed image and the ground truth as the loss function. The Adam [34] optimizer is used with beta1=0.9 and beta2=0.999. All the parameters are initialized with He initialization and the whole network is trained end-to-end. Following [6], the learning rate starts with $1e - 4$ for all modules and decays in half every 200 epochs. We parallelly run our ITSRN-RDN on two GeForce GTX 1080Ti GPU with mini-batch size 16 and it cost 2 days to reach convergence (about 500 epochs). During the test, we directly feed the whole LR image into our network (as long as the graphic memory is sufficient) to generate the SR result.

## 5.3 Comparison with State-of-the-arts

To demonstrate the effectiveness of the proposed SR strategy, we compare our method with state-of-the-art continuous SR methods, *i.e.,* MetaSR [5] and LIIF [6]. We also compare with the discrete SR methods, *i.e.,* RDN [18] and RCAN [4], which are state-of-the-art natural image SR methods. Since RDN and RCAN rely on specific up-sampling modules, they have to train different models for different upsampling scales and cannot be tested for the scales not in the training set. For a fair comparison, we retrain all the compared methods using our training set with the recommended parameters and codes released by the authors. Note that, during test, we downsample the ground truth with different ratios to generate the LR inputs for different magnification ratios. For larger magnification ratios, the details are fewer in its corresponding LR input.

Table 1: Quantitative comparison on SCI1K and SCI1K-compression test sets in terms of PSNR (dB). The best (second best) results are in red (blue). RDN [18] and RCAN [4] use different models for different upsampling scales. MetaSR [5], LIIF [6] and ITSRN(ours) use one model for all the upsampling scales, and the three models are trained with continuous random scales uniformly sampled from ×1 ∼ ×4.

| Method | Dataset: SCI1K | | | | | | Dataset: SCI1K-compression | | | | | |
| | In-training-scale | | | Out-of-training-scale | | | In-training-scale | | | Out-of-training-scale | | |
| | ×2 | ×3 | ×4 | ×5 | ×7 | ×9 | ×2 | ×3 | ×4 | ×5 | ×7 | ×9 |
|---|---|---|---|---|---|---|---|---|---|---|---|---|
| Bicubic [18] | 28.81 | 25.15 | 23.18 | 22.02 | 20.72 | 19.96 | 28.28 | 24.87 | 22.99 | 21.84 | 20.58 | 19.84 |
| RDN [18] | 38.45 | 33.59 | 29.81 | - | - | - | 35.16 | 30.60 | 27.17 | - | - | - |
| RCAN [4] | 38.61 | 33.91 | 30.80 | - | - | - | 35.25 | 31.15 | 27.78 | - | - | - |
| MetaSR-RDN [5] | 38.57 | 33.67 | 30.12 | 27.52 | 23.91 | 22.02 | 35.20 | 30.96 | 27.63 | 25.31 | 22.57 | 21.30 |
| LIIF-RDN [6] | 38.65 | 33.97 | 30.55 | 27.77 | 23.99 | 22.18 | 35.43 | 31.07 | 27.69 | 25.27 | 22.59 | 21.36 |
| ITSRN-RDN(Ours) | 38.74 | 34.32 | 30.82 | 28.15 | 24.36 | 22.36 | 35.53 | 31.31 | 28.02 | 25.62 | 22.79 | 21.45 |

Table 2: Quantitative evaluation on SCI quality assessment datasets in terms of PSNR (dB). The best (second best) results are in red (blue). RDN [18] and RCAN [4] train different models for different upsampling scales. The rest methods train one model for all the upsampling scales. All the models are trained on the SCI1K training set.

| Dataset | Method | In-training-scale | | | Out-of-training-scale | | | | | |
| | | ×2 | ×3 | ×4 | ×5 | ×6 | ×7 | ×8 | ×9 | ×10 |
|---|---|---|---|---|---|---|---|---|---|---|
| SCID [12] | RDN [18] | 34.00 | 28.34 | 25.74 | - | - | - | - | - | - |
| | RCAN [4] | 33.90 | 28.98 | 26.02 | - | - | - | - | - | - |
| | MetaSR-RDN [5] | 33.84 | 29.08 | 25.76 | 23.62 | 22.38 | 21.59 | 21.07 | 20.71 | 20.41 |
| | LIIF-RDN [6] | 34.24 | 29.10 | 25.89 | 23.77 | 22.53 | 21.73 | 21.21 | 20.84 | 20.54 |
| | ITSRN-RDN | 34.19 | 29.46 | 26.22 | 23.96 | 22.64 | 21.80 | 21.26 | 20.87 | 20.56 |
| SIQAD [13] | RDN [18] | 33.53 | 26.89 | 23.38 | - | - | - | - | - | - |
| | RCAN [4] | 32.87 | 27.27 | 23.69 | - | - | - | - | - | - |
| | MetaSR-RDN [5] | 34.12 | 28.40 | 23.55 | 21.18 | 20.18 | 19.63 | 19.25 | 18.94 | 18.65 |
| | LIIF-RDN [6] | 34.31 | 28.27 | 23.44 | 21.16 | 20.25 | 19.70 | 19.36 | 19.02 | 18.70 |
| | ITSRN-RDN | 34.68 | 29.07 | 24.03 | 21.44 | 20.38 | 19.77 | 19.40 | 19.09 | 18.79 |

Table 1 and Table 2 present the quantitative comparison results on different datasets. Table 1 lists the average SR results for 200 screen content images in our SCI1K and SCI1K-compression test sets. All the methods are retrained on the corresponding training sets of SCI1K and SCI1K-compression. It can be observed that our method consistently outperforms all the compared methods. Compared with RDN, which is our backbone, our method achieves nearly 1 dB gain at ×4 SR on SCI1K test set. Compared with the competitive RCAN, our method still achieves nearly 0.4 dB gain at ×3 upsampling for the uncompressed dataset. Note that at ×4 SR, the result of our method is similar as that of RCAN. The main reason is that our backbone RDN generates much worse results than RCAN at ×4 SR. If we change our backbone to more powerful structures, our results could also be further improved. For the scales that are not used in the training process (denoted by out-of-training-scales), RDN and RCAN are not applicable. Meanwhile, our method outperforms the continuous SR methods (*i.e.,* MetaSR and LIIF), which demonstrates that the proposed implicit transformer scheme is superior in modeling both coordinates and features. Table 2 presents the SR results on two SCI quality assessment datasets. Since the images in the two datasets are not compressed, we directly utilize the models trained on SCI1K training set to test. It can be observed that our method still outperforms the compared methods.

Besides visual results in Fig. 1, Fig. 3 presents an example of the qualitative comparison results on SCI1K test set[3]. It can be observed that our method recovers more realistic edges of characters than the compared methods. Fig. 4 presents the visual results for ×10 SR. It can be observed that our method reconstructs the thin edges better than the compared methods at large magnification ratios.

---

[3]More visual comparison results are presented in the supplementary file.

Table 3: Ablation study for scale token and implicit position encoding. The PSNR/SSIM results are the averaging results on all the SCI1K test images.

| Scale token | | $\times$ | $\checkmark$ | $\times$ | $\checkmark$ |
|---|---|---|---|---|---|
| implicit position encoding | | $\times$ | $\times$ | $\checkmark$ | $\checkmark$ |
| In-training-scale($\times$4) | PSNR | 30.43 | 30.60 | 30.76 | 30.82 |
| | SSIM | 0.9329 | 0.9351 | 0.9353 | 0.9364 |
| Out-of-training-scale($\times$6) | PSNR | 25.94 | 25.95 | 26.05 | 26.00 |
| | SSIM | 0.8686 | 0.8715 | 0.8725 | 0.8746 |

## 5.4 Ablation Study Results

In this section, we perform ablation study to demonstrate the effectiveness of the proposed modules. Table 3 lists quantitative comparison results on SCI1K test set. It can be observed that the scale token and implicit position encoding totally contributes 0.39 dB to the final SR result at $\times$4 SR. Even for the out-of-training-scales, the two modules also contribute 0.06 dB gain. Note that, the gain is lower for larger magnification factor is because that the thin edges in screen contents are hardly to be distinguished after $\times$6 downsampling. Therefore, it is difficult to bring gains with scale token and position encoding.

Table 4: Comparison of using different realizations of the weight $w$ in IPE. "Fixed" means $w$ is calculated based on the spatial distance between the query and its neighbors."Learned" refers to learning $w$ via an MLP, which is used in this work.

| Weight $w$ in IPE | In-training-scale | | | Out-of-training-scale | | | | |
|---|---|---|---|---|---|---|---|---|
| | $\times$2 | $\times$3 | $\times$4 | $\times$5 | $\times$7 | $\times$9 | $\times$24 | $\times$40 |
| Fixed | 35.27 | 31.03 | 27.78 | 25.47 | 22.78 | 21.45 | 18.42 | 17.31 |
| Learned | 35.53 | 31.31 | 28.02 | 25.62 | 22.79 | 21.45 | 18.40 | 17.29 |

Table 5: Comparison of SR performance with different training scales.

| Scale | $\times$4 | $\times$6 | $\times$8 | $\times$10 |
|---|---|---|---|---|
| Training scale $\times$2-$\times$4 | 30.82 | 26.00 | 23.16 | 21.77 |
| Training scale $\times$2-$\times$8 | 30.83 | 26.37 | 23.59 | 21.86 |

We also conduct ablation study on the $w$ in IPE by changing it to different realizations. Table 4 lists the SR results on SCI1K-compression test set. From the table, we can observe that our proposed learnable scheme is superior to the fixed scheme when the input LR images suffer from JPEG compression. This is because that, with JPEG compression, the neighboring information can help alleviate the compression artifacts and the fixed distance based weighting strategy is not suitable in this case. Note that, for very large magnification ratios, the learned $w$ is slightly inferior to the fixed $w$ by 0.02 dB since the heavy distortions in the LR input may confuse the learning process. Besides, we conduct ablation study on the training scales. As shown in Table 5, the SR results at $\times$6 and $\times$8 upsampling with training scales $\times$2 $-$ $\times$8 are better than that with training scale $\times$2 $-$ $\times$4. The main reason is that the scales of $\times$6 and $\times$8 are in the range of the second training scales. In addition, the second strategy is also beneficial for $\times$10 upsampling. This indicates us that using wide distributed training scales is better than using narrow distributed training scales.

## 5.5 Discussion

The most related work to our proposed ITSRN is LIIF [6], which utilizes implicit function for continuous SR. However, it directly concatenates the image coordinate and its CNN feature, and then map them to the RGB value via an MLP. Different from it, we reformulate the SR process as an implicit transformer. There are two MLPs in our scheme. The first MLP is utilized to model the relationship between pixel coordinates. Then the "relationship" is multiplied by the pixel features to generate the pixel value, which can also be termed as an attention strategy. The second MLP is utilized for implicit position encoding, which is useful to keep the continuous of the reconstructed

pixels. The experiment results also demonstrate that the proposed two strategies make our method consistently outperform LIIF on all the four test sets.

# 6 Conclusion

In this paper, we propose a novel arbitrary-scale SR method for screen content images. With the proposed implicit transformer and implicit position encoding modules, the proposed method achieves the best results on four datasets at various magnification ratios. Due to the continuous magnification ability, our method enables users to display the received screen contents on screens with various sizes. In addition, we construct the first SCI SR dataset, which will facilitate more research on this topic.

Note that, although our method outperforms state-of-the-arts on screen content images, it may not work the best for natural images at a fixed magnification ratio. The main reason is that our method is designed for SCIs with high contrast and dense edges, which are suitable to be modeled by point-to-point mapping. Besides, we simulate the degradation process with bicubic and JPEG compression, which may be different from the actual degradation in transmission and acquisition. Thus, there will be limitation in practical applications. In the future, we would like to develop blind distortion based SCI SR to make our model adapt to real scenarios better.

## Acknowledgments

The authors would like to thank the anonymous reviewers for their valuable comments. This research was supported in part by the National Natural Science Foundation of China under Grant 62072331, Grant 62171317, and Grant 61771339.

## Broader Impact

This work is an exploratory work on screen content images super-resolution with arbitrary-scale magnification. The constructed dataset can facilitate research on this topic. In addition, the proposed method can be combined with image (video) compression technology to enable screen contents transmission with limited bandwidth. As for societal influence, this work will improve the quality of pictures displayed on the screen of any resolution. However, We would like to point out that SR is actually predicting (hallucinating) new pixels, which may make the image deviate from the ground truth. Therefore, image SR has a weak link with deep fakes. It is worth noting that the positive social impact of this technology far exceeds the potential problems. We call on people to use this technology and its derivative applications without harming the personal interests of the public.

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
