# Implicit Transformer Network for Screen Content Image Continuous Super-Resolution (Supplementary Materials)

**Jingyu Yang**[1]    **Sheng Shen**[1]    **Huanjing Yue**[1]*    **Kun Li**[2]
[1]School of Electrical and Information Engineering, Tianjin University
[2]College of Intelligence and Computing, Tianjin University
{yjy, codyshen, huanjing.yue, lik}@tju.edu.cn
https://github.com/codyshen0000/ITSRN

## A    Appendix

### A.1    Datasets

Four datasets are involved in our experiments, *i.e.* SCI1K, SCI1K-compression, SCID, and SIQAD. Among them, SCID and SIQAD are public available datasets, which can be downloaded from their urls[2][3]. Our constructed datasets, *i.e.* SCI1K and SCI1K-compression, will be public available after the acceptance of this work. Fig. 1 presents some examples of our dataset. It can be observed that our dataset is constructed with various screen contents, such as documents, slides, gaming scenes, and cartoons. All the four datasets do not contain personally identifiable information or offensive content.

### A.2    Visual Comparison Results

In this section, we present more visual results by comparing with RDN [1], RCAN [2], MetaSR [3], and LIIF [4]. Fig. 2 presents the arbitrary SR results. It can be observed that the proposed method outperforms state-of-the-art continuous SR methods. The LR input only contains $36 \times 36$ pixels, and the characters are hardly to be identified. After SR reconstruction, our method reconstructs the sharp and thin edges of characters, which makes them be easily recognized.

Fig. 3 presents the visual comparison results at $\times 4$ SR. Our method outperforms both the discrete SR methods and continuous SR methods. Besides characters, our method also works the best in dealing with dense edge structures.

Fig. 4 presents the visual comparison results for the compressed images. As introduced in the main paper, the HR image is first downsampled with a specific scale factor to generate the clean LR image, and then the clean LR is compressed into JPEG image with a specific quality factor(QF) via PIL (Python Imaging Library) JPEG encoder. In this way, we can simulate the degradation introduced in transmission and storage for screen content images. As shown in Fig. 4, although all the methods are retrained with the SCI1K-compression dataset, our method is more robust to the compression distortion. One reason is that we introduce a learnable weighting factor in implicit position encoding.

---

*Corresponding author: Huanjing Yue
[2]SCID: http://smartviplab.org/pubilcations/SCID/SCID.zip
[3]SIQAD: https://sites.google.com/site/subjectiveqa/

35th Conference on Neural Information Processing Systems (NeurIPS 2021).

Table 1: Comparison of ×4 SR performance for natural images.All the models(except TTSR) are trained on DIV2K dataset and test on Urban100 and B100 dataset.

| Methods | EDSR[6] | SRGAN[7] | RDN[1] | RCAN[2] | SRFBN[8] | TTSR[9] |
|---|---|---|---|---|---|---|
| Years | CVPR2017 | CVPR2017 | CVPR2018 | ECCV2018 | CVPR2019 | CVPR2020 |
| Urban100 | 26.64 | – | 26.61 | 26.82 | 26.60 | 25.87 |
| B100 | 27.71 | 25.16 | 27.72 | 27.77 | 27.72 | - |
| Methods | TTSR*[9] | IGNN[10] | NLSN[11] | MetaSR[3] | LIIF[4] | Our |
| Years | CVPR2020 | NeurIPS2020 | CVPR2021 | CVPR2019 | CVPR2021 | - |
| Urban100 | 25.26 | 26.84 | 26.96 | 26.55 | 26.68 | 26.73 |
| B100 | 27.27 | 27.77 | 27.78 | 27.71 | 27.74 | 27.72 |

Table 2: Comparison of Flops, Inference Time, Memory, and Parameter for $128 \times 128$ LR input with ×4 upsampling.

| Methods | TTSR | EDSR | RCAN | RDN | LIIF | Our |
|---|---|---|---|---|---|---|
| GFlops(G) | 409 | 824 | 261 | 373 | 723 | 1032 |
| Infertime(s) | 0.05 | 0.02 | 0.15 | 0.05 | 0.12 | 0.09 |
| Memory(M) | 4723 | 1202 | 3755 | 1267 | 1471 | 4579 |
| Parameters(M) | 6.73 | 43.09 | 15.59 | 22.27 | 22.32 | 22.61 |

## A.3 Evaluation on Natural Images

To further evaluate the effectiveness of our proposed method on natural images, we retrain our model on the widely used DIV2K dataset, and evaluate it on urban100 and B100 datasets. Tabel 1 presents the results of our method and state-of-the-art natural image SR methods at ×4 upsampling. Since TTSR is a transformer based SR network, we also present the original TTSR results on the two datasets by quoting from their paper (this model is trained on CUFED5 [5]). In addition, for a fair comparison, we retrain TTSR on the DIV2K dataset and the corresponding model is denoted as TTSR*. The PSNR value is calculated on Y channel in the transformed YCbCr space. It can be observed that, for natural image dataset Urban100, our method still outperforms the continuous SR method LIIF. On the B100 dataset, our method is only 0.06 dB less than the best method NLSN. It demonstrates that although our method is not the best for natural image SR, but still achieves promising results on natural images.

## A.4 Complexity Analysis

Table 2 lists the computation costs and inference times of our method and compared methods by evaluating with an Nvidia GTX 1080Ti. Regarding the inference time, our model is faster than RCAN and LIIF. Our memory cost is less than that of TTSR. Our memory cost is large since our model contains point-wise MLP structures. In the future, we will optimize the MLP structure to reduce the memory cost and FLOPs.

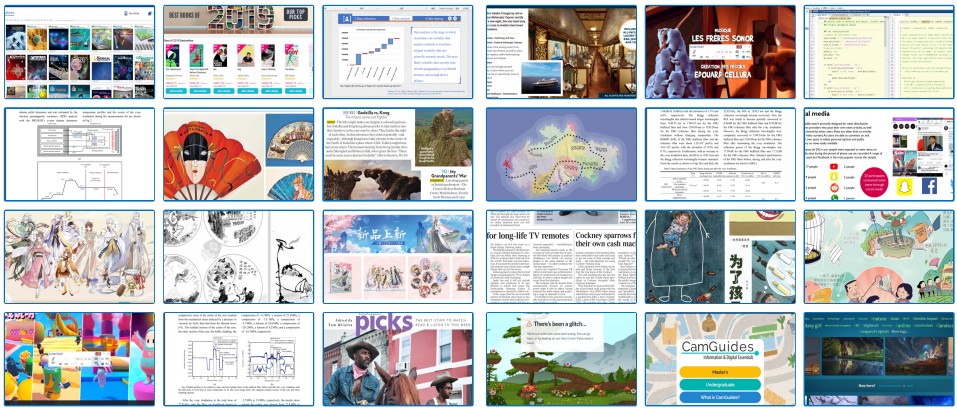

Figure 1: Sample images in our SCI1K dataset.

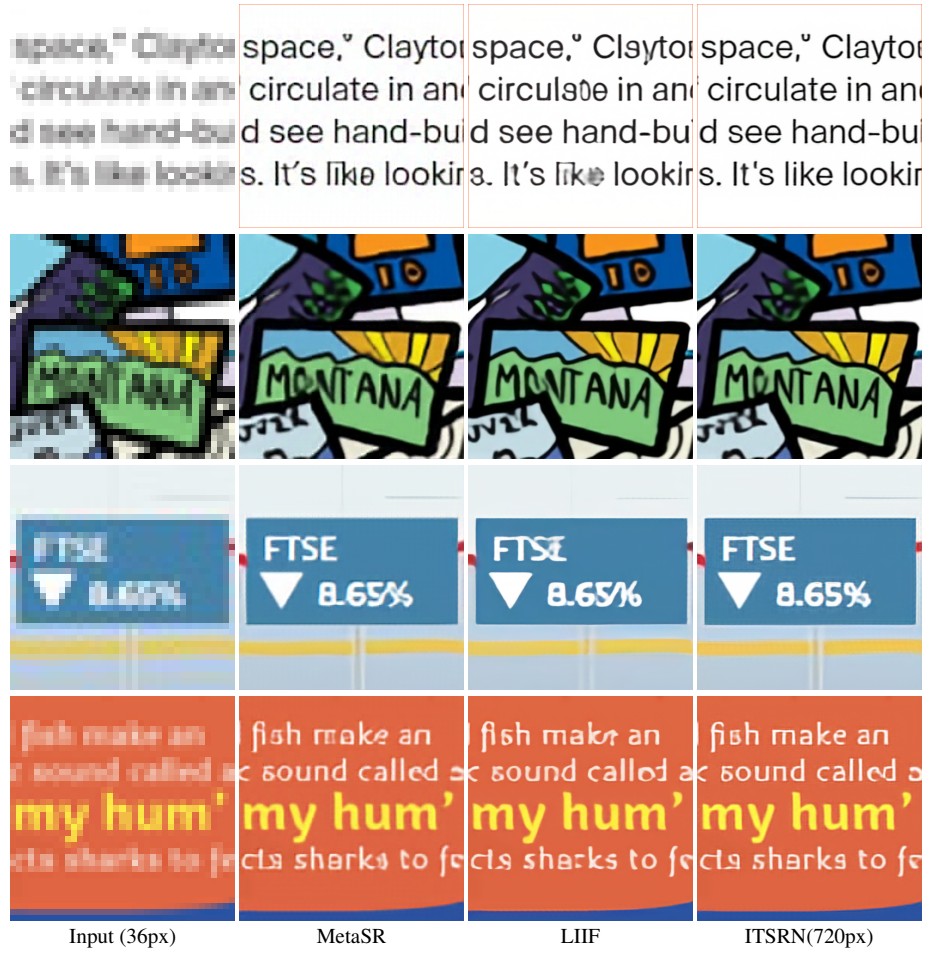

| Input (36px) | MetaSR | LIIF | ITSRN(720px) |

Figure 2: Visual comparison with state-of-the-arts for arbitrary SR results. The input is a $36 \times 36$ patch from an image in SCI1K test set. All the three models are trained with continuous scales in the range $\times 1 \sim \times 4$ and are tested for $\times 20$ magnification. Note that the LR input is generated with $\times 4$ downsampling, and there is no ground truth for its $\times 20$ magnification.

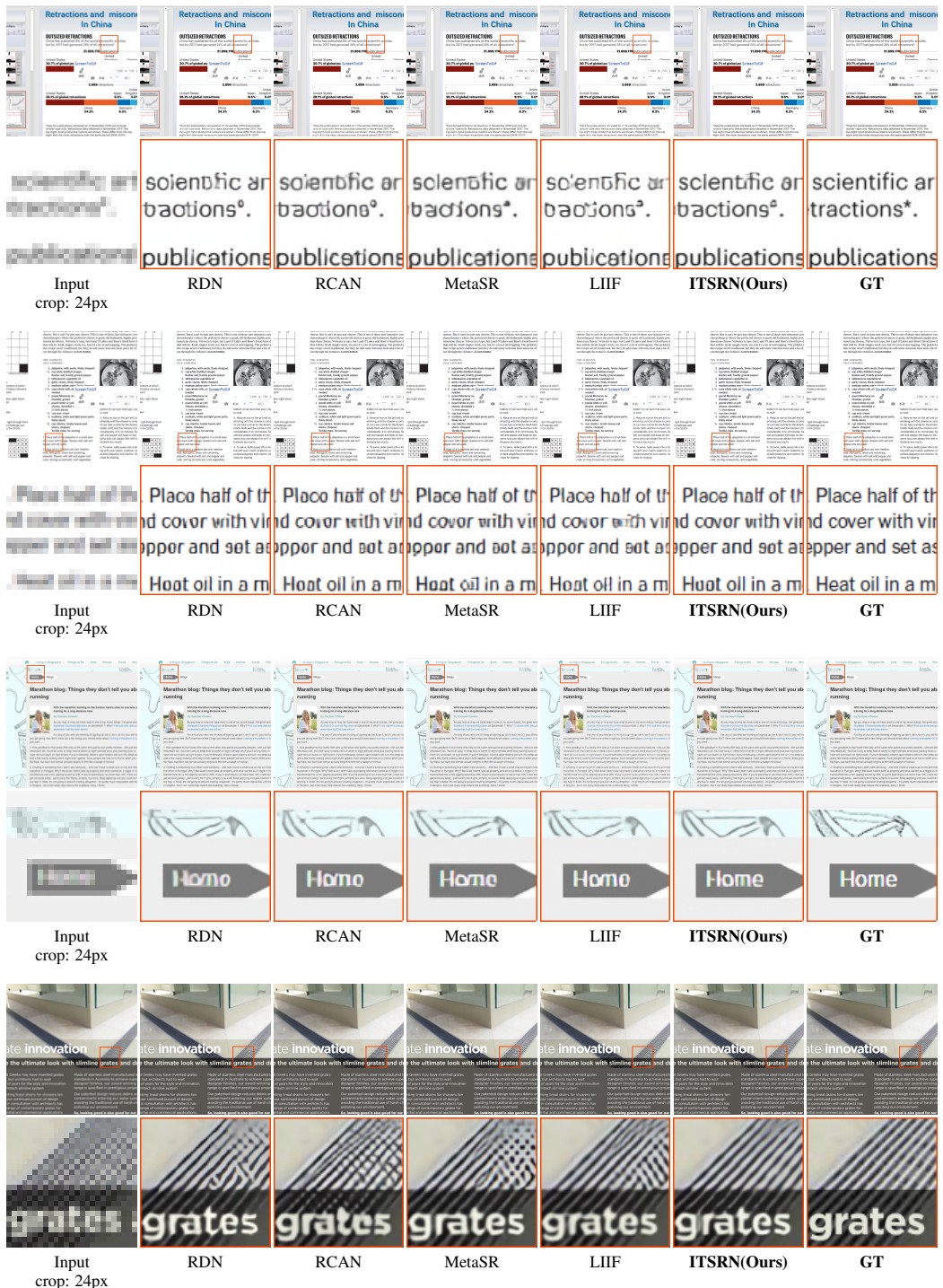

Figure 3: Visual comparison with state-of-the-arts at ×4 SR. The cropped patch size is given at the bottom of the LR input.

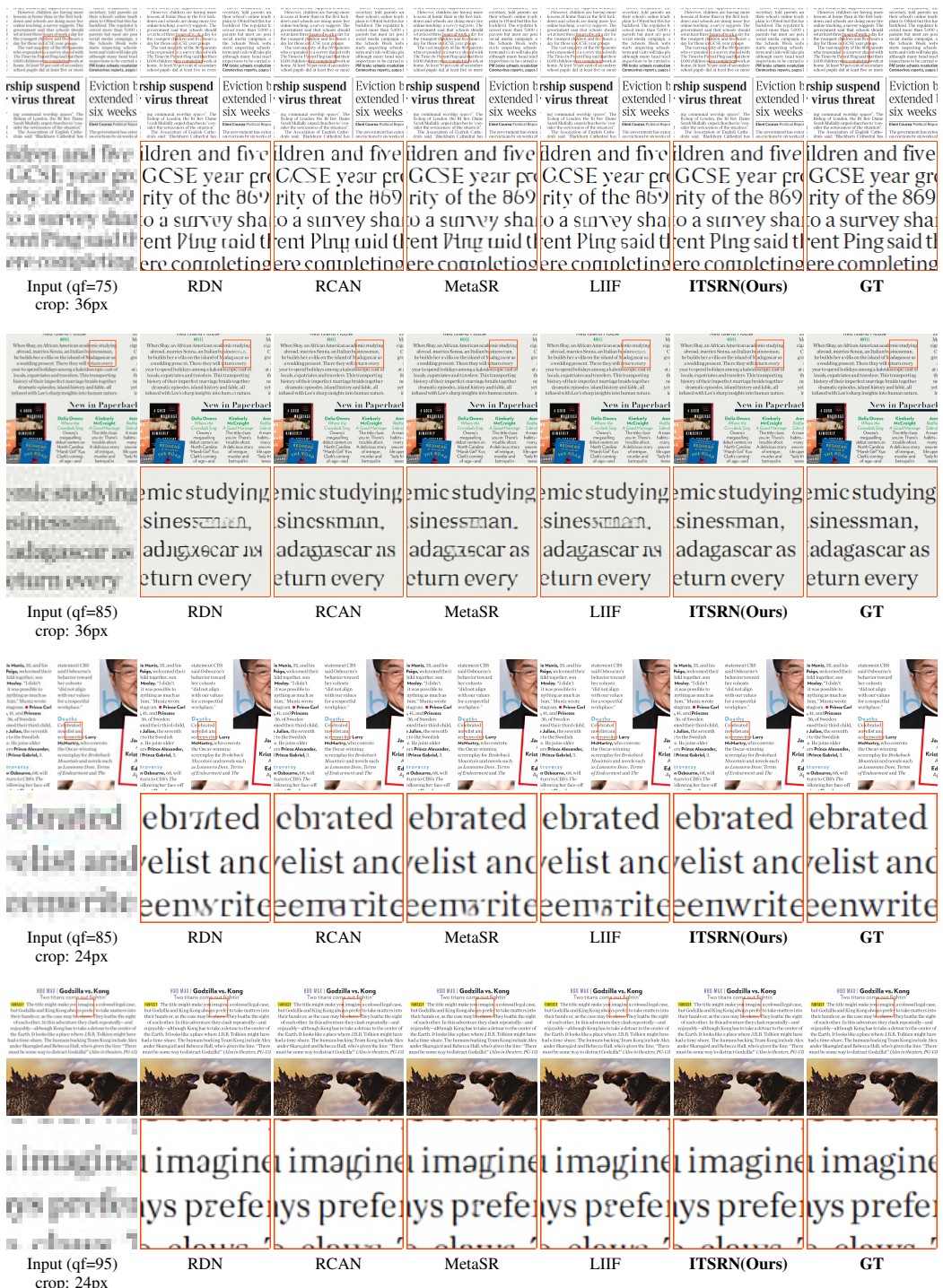

Figure 4: Visual comparison with state-of-the-arts at ×4 SR on SCI1K-compression dataset. The quantization factor (qf) used for LR image generation and the cropped patch size are given at the bottom of the input image.