# OpenReview forum: "Implicit Transformer Network for Screen Content Image Continuous Super-Resolution"
_NeurIPS.cc/2021/Conference — NeurIPS 2021 Poster_

### Official Review · Reviewer_zDGM · 2021-07-12

**Rating:** 7
**Confidence:** 4

**Summary:**

This paper presents a novel method called implicit transformer for the task of super-resolution of screen content images. The proposed method is able to integrate the advantage of transformer in capturing relation between pixels and that of implicit neural representation in producing continuous and sharp results. The effectiveness of the proposed method is validated on both newly collected datasets w/o and w/ compression degradation, and datasets originally created for image quality assessment. Ablation study also demonstrates the role of the proposed modules.

**Limitations And Societal Impact:**

Yes, they have addressed the limitation and potential negative societal impact.

**Main Review:**

Overall, the idea of integrating implicit neural representation into a transformer is interesting and its effectiveness has been validated in the extensive ablation study and comparison with SOTA methods. However, there are still some issues to be clarified.
1.	It is not clear what is the meaning of computing relation matrix using coordinates in transformer. Since coordinates are content-independent, it means that for images of the same size, they share the same $\phi(q, k)$.
2.	In the comparison, the proposed method shows better performance on both qualitative and quantitative comparison. However, it is not clear whether comparison is fair since number of parameters and flops are not mentioned.

**Time Spent Reviewing:**

2.5 hours

---

> ### Author Response · Authors · 2021-08-09
> **Response to Reviewer#5 zDGM**
>
> Thanks for your nice summary of our work and positive comments on our method, experiments and datasets! In the following, we give detailed response to your comments one by one.
>
> >*It is not clear what is the meaning of computing relation matrix using coordinates in transformer. Since coordinates are content-independent, it means that for images of the same size, they share the same ϕ(q,k)*
>
> Reply: Yes, the relation ϕ(q,k) is the same for the images of the same size if they share the same upsampling scale. For images with the same size but different magnification ratios, ϕ(q,k) is different since the scale token is involved for the nonlinear mapping process, as shown in Eq. (5). Since the features $v$ for each position is different, we obtain the spatial adaptive pixel value $I_q$ via Eq. (2). In addition, we utilize implicit position encoding, as shown in Eq. (7), to further refine the pixel values according to its neighboring pixel values. We would like to point out that if the input size and magnification factor are both fixed, during test, we can skip the computation of the relationship matrix ϕ(q,k) and directly utilize a pre-computed one. In this way, our flops and parameters can be greatly reduced.
>
> >*In the comparison, the proposed method shows better performance on both qualitative
> and quantitative comparison. However, it is not clear whether comparison is fair since
> number of parameters and flops are not mentioned.*
>
> Reply: Thanks for your suggestion. Table R1 presents the computing complexity of our method and compared methods. It can be observed that our inference time is faster than RCAN and LIIF. Our memory cost is less than that of TTSR. In the future, we would like to optimize the MLP structure to reduce our memory cost and Flops.
>
> Table R1. Comparison of Flops, Inference Time, Memory, and #Parameter for $128\times128$ LR input with $4\times$ upsampling.
>
> ||TTSR| EDSR | RCAN |RDN  | LIIF | Our |
> |---|---| --- | --- | --- | --- | --- |
> |GFlops|409 | 824|  261 |  372.7| 723 | 1032 |
> |Inference Time| 0.05s|0.02s | 0.15s |  0.05| 0.12s | 0.09s |
> |Memory| 4723M |1202M| 3755M | 1267M |1471M  | 4579M |
> |Parameters| 6.73M|43.09M|15.59 M |22.27M |22.32M | 22.61M|

---

> > ### Comment · Reviewer_zDGM · 2021-08-25
> > **Comments after rebuttal**
> >
> > Authors have addressed my concerns, so I keep my original score of Accept.

---

### Official Review · Reviewer_vvmM · 2021-07-16

**Rating:** 7
**Confidence:** 4

**Summary:**

This work presents a continuous image super-resolution based on an implicit transformer. Authors extend LIIF [6] with some modifications including transformer-like formulation in (2), scale token in (5), and implicit position encoding (7). The proposed implicit transformer learns the transformation weight from query (high-resolution) coordinate, key (low-resolution) coordinate, and scale token. A pixel intensity is then computed by transforming the pixel features (obtained from CNN backbone) with the transformation weights. The pixel intensity is further refined with the implicit position encoding, which is an adaptive weighting scheme with learned weights. Experimental results on screen content images demonstrate superior performance over existing continuous super-resolution methods [5,6].

**Limitations And Societal Impact:**

The authors did not address the potential negative societal impact of their work.

**Main Review:**

Pros
1) The implicit transformer with scale tokens seems to be an appealing solution for continuous super-resolution.
2) The results on screen content images including sharp and thin edges are excellent.
3) Intensive ablation study is provided.

Cons
1) A proper motivation of the proposed methods is needed.
- The transformer-like formulation in (2) is presented to avoid the issue by a direct concatenation of pixel features and relative coordinates in LIIF [6]. However, it seems that solid justification about (2) is rather insufficient.
- L151-152: Why did you choose the subtraction operation?
- (7) is an adaptive weight scheme in which the weight $w$ is learned from MLP. 'Implicit position encoding' does not seem like the right term.

2) RDN [18] and RCAN [4] are too out-of-date to be compared.

3) The comparison was conducted with only screen content images containing sharp and thin edges (e.g., text) yet less textures. However, some screen contents may also have diverse textures like natural images.
How is the performance in the natural image? It would be better to provide the performance evaluation in the natural images in order to clarify the limitation of this work, if the performance on the natural image is not as good as that of state-of-the-art SR methods (including both discrete and continuous SR approaches).

4) Some minor issues
- L145: $I_q$ is computed with the multiplication of $\phi$ and $v$, not the weighted average of $v$.
- L171: 'implicit means' is a wrong expression.


**Time Spent Reviewing:**

6 hours

---

> ### Author Response · Authors · 2021-08-09
> **Response to Reviewer#4 vvmM**
>
> Thanks for your nice summary and positive comments on our method and results! In the following, we give detailed response to your comments one by one.
>
> > *A proper motivation of the proposed methods is needed.
>  a) The transformer-like formulation in (2) is presented to avoid the issue by a direct concatenation of pixel features and relative >coordinates in LIIF [6]. However, it seems that solid justification about (2) is rather insufficient.
>  b) L151-152: Why did you choose the subtraction operation?
> c) (7) is an adaptive weight scheme in which the weight w is learned from MLP. 'Implicit position encoding' does not seem like the right term.*
>
> Reply: Thanks for your careful review.
>
> 1. Image SR is essentially to interpolate new pixels with the information of its neighboring pixels. The Eq. (2) ($\phi(q,k)v$) is utilized to model the relationship between pixel coordinates. Then the “relationship” is multiplied by the pixel features to generate the pixel value, which can also be termed as an attention strategy. While in LIIF, it utilizes $\phi(q,k,v)$ to directly generate the pixel value $I_q$. In other words, we add additional constraints to the relationship between $v$ and $\phi(q,k)$ while there is no constraints in LIIF. In this way, the task of our network is more specific than that of LIIF and the solution space is more specific than that of LIIF. As demonstrated in previous CNN networks, introducing constraints in the network structure is beneficial for the final performance. For example, ResNet uses skip connections to constrain the network to learn residuals, which is greatly better than directly mapping. In addition, we perform an ablation study by replacing the proposed $\phi(q,k)v$ with $\phi(q,k,v)$, i.e. the one proposed in LIIF, and keep the other modules the same as that in our ITSRN. Table R1 presents the comparison results. It can be observed that our module greatly outperforms that proposed in LIIF.
>
> Table R1. Comparison between the mapping strategy $\phi(q,k)v$ and $\phi(q,k,v)$ evaluated on  SCI1K dataset.
>
> | SCI1K | x4 | x6 | x8 |
> | --- | --- | --- | --- |
> | $\phi(q,k)v$|30.82  | 26.00  | 23.16 |
> | $\phi(q,k,v)$|30.34 |  25.77 | 23.00 |
>
> 2. Subtraction is an efficient way to model the relationship between pixel coordinates. The dot product and Hadamard product is reasonable in modeling the relationship between features. However, our target is estimating the relationship between coordinates. Therefore, we utilize subtraction, which also has the least computation cost.
> 3. Yes, (7) is an adaptive weight scheme. We utilize ‘implicit position encoding’ because (7) also models the relationship between neighboring pixels. In [R7], the authors claimed that convolution, which is also modeling the relationship between neighboring pixels, could be also considered as an implicit position encoding scheme. Therefore, we termed (7) as implicit position encoding. We will make this point clear in the final version.
>
> >*RDN [18] and RCAN [4] are too out-of-date to be compared.*
>
> Reply: We compare with RDN[18] because our feature extraction backbone is RDN. Although RCAN[4] was published in 2018, its performance is still state-of-the-art. As shown in Table R2, the performance of RCAN was still on the top three for natural image SR. Since retraining many state-of-the-art methods on our dataset in the rebuttal period is not feasible, we compare with 10 benchmark SR methods on natural image SR datasets B100 and Urban100. The PSNR results for compared methods are quoted  from their papers (using the same training dataset). As shown in Table R1, even for natural images, our model still provides promising results. In the final version, we will retrain more SOTA SR methods on our dataset to demonstrate the superiority of the proposed method for screen content SR.
>
> Table R2. Comparison of $\times4$ SR performance for natural images. All the methods (*except TTSR*) are trained on DIV2K dataset and test on Urban100 and B100 datasets.
>
> | Methods |  EDSR[R1] | SRGAN[R2] | RDN | RCAN |SRFBN[R3]  |
> | --- | --- | --- | --- | --- |  --- |
> |  Years| CVPR2017 | CVPR2017 | CVPR2018 | ECCV2018  | CVPR2019  |
> | Urban100 |26.64  | ---- | 26.61 | 26.82 | 26.60 |
> | B100 |27.71  | 25.16 | 27.72 | 27.77 | 27.72 |
>
> | Methods |  TTSR[R4] | IGNN[R5] | NLSN[R6] | MetaSR | LIIF  | Our |
> | --- | --- | --- | --- | --- |  --- | --- |
> | Years| CVPR2020 | NeurIPS2020 | CVPR2021 | CVPR2019 | CVPR2021 ||
> | Urban100 |25.87  |26.84|  26.96 |26.55 |26.68 | 26.73 |
> | B100 |---- |27.77| 27.78 | 27.71| 27.74| 27.72 |
>
> >*The comparison was conducted with only screen content images containing sharp and thin edges (e.g., text) yet less textures. However, some screen contents may also have diverse textures like natural images. How is the performance in the natural image? It would be better to provide the performance evaluation in the natural images in order to clarify the limitation of this work, if the performance on the natural image is not as good as that of state-of-the-art SR methods (including both discrete and continuous SR approaches).*
>
> Reply: Thanks for your suggestion. As mentioned in the reply to Q2, we evaluate our method on natural image datasets by retraining our method with DIV2K dataset. As shown in Table R2, our method is only 0.06 dB less than the best method NLSN on the B100 dataset. We will add this result in the final version.
>
> >*Some minor issues
> L145: Iq is computed with the multiplication of ϕ and v, not the weighted average of v.
> L171: 'implicit means' is a wrong expression.*
>
>
> Reply: Thanks for your careful review. We will correct these issues in the final version.
>
> ><I>**Limitations And Societal Impact**: The authors did not address the potential negative societal impact of their work.</I>
>
> Reply: As suggested by Reviewer#3  2ePa , we will add the negative societal impact, *i.e*., deep fakes, in the final version.
>
>
> [R1] Bee Lim, Sanghyun Son, Heewon Kim, Seungjun Nah, and Kyoung Mu Lee, "Enhanced Deep Residual Networks for Single Image Super-Resolution," 2nd NTIRE: New Trends in Image Restoration and Enhancement workshop and challenge on image super-resolution in conjunction with CVPR 2017.
>
> [R2] Ledig C, Theis L, Huszár F, et al. Photo-realistic single image super-resolution using a generative adversarial network, CVPR, 2017: 4681-4690.
>
> [R3] Li Z, Yang J, Liu Z, et al. Feedback network for image super-resolution, CVPR, 2019: 3867-3876.
>
> [R4] Yang F, Yang H, Fu J, et al. Learning texture transformer network for image super-resolution, CVPR, 2020: 5791-5800.
>
> [R5] Zhou, Shangchen, et al. “Cross-Scale Internal Graph Neural Network for Image Super-Resolution.” NeurIPS, vol. 33, 2020, pp. 3499–3509.
>
> [R6] Mei, Yiqun, et al. “Image Super-Resolution With Non-Local Sparse Attention.” CVPR, 2021, pp. 3517–3526.
>
> [R7] Islam, Md Amirul, Sen Jia, and Neil DB Bruce. "How much position information do convolutional neural networks encode?" ICLR, 2020.

---

> > ### Comment · Reviewer_vvmM · 2021-08-23
> > **Comments on the rebuttal**
> >
> > Authors successfully addressed most of the concerns in the rebuttal, so I changed the review rating from '6: Marginally above the acceptance threshold' to '7: Good paper, accept'.

---

### Official Review · Reviewer_2ePa · 2021-07-16

**Rating:** 5
**Confidence:** 4

**Summary:**

This work addresses the SCISR problem by combining transformers to the network architecture. The backbone of the architecture utilizes a CNN to extract feature maps, to enlarge the receptive field, an implicit transformer is used. After that the implicit position encoding is conducted via MLP. Natural images are suitable for SR but there is a lack of SR techniques that are suitable for screen content, which this paper tries to address.


**Ethical Concerns:**

See above

**Limitations And Societal Impact:**

SR has the potential to be a piece of technology that can impact the society perhaps in a negative way. SR in the end is trying to construct new pixels or even hallucinate details that were not seen or non existent. This means that the picture may not be entirely true to the true visual. This has a weak link with deep fakes. Some thought on this is required. However, the authors do not address such societal or ethical implications in this paper.


**Main Review:**

The method is described well. The experiments show that the method can perform well in terms of quantitative analysis and qualitative comparison. The qualitative comparison shows that the method is well suited for screen content compared to other SOTA methods.
With the help of transformers, the SR seems to improve in quality, but there is no analysis on computational complexity. Normally, when the transformer is used, it can be utilized to reduce the computational load while also improving quality. Does this work also convey such characteristics?
The latest SOTA for SR using transformers is the CVPR 2020 paper TTSR which is not mentioned in the main paper. There is no comparison with the method either nor in the related works section. The latest SOTA is essential to show in the paper so that it can be compared to the latest SOTA. ALthough this work is focussed on SCISR, it is still very related to SR and further uses transformers as a component to the network architecture. The most similar work is arguably the TTSR which need s to be addressed.
Also, the authors conduct experiments on RDN, RCAN, MetaSR etc. However, these are somewhat outdated SR models. Thus, it is not entirely surprising that the current model outperforms them. The latest SR models that the authors should consider is definitely TTSR, but also EDVR, EDSR, IGNN, SRGAN etc.
I would say the experimental section is not as extensive as I would have expected. Although there are not many methods combining transformers with CNNs yet for SR, it is much more relevant to compare with CNN only methods to show the effectiveness for combining an additional module; the transformer.


**Time Spent Reviewing:**

6

---

> ### Author Response · Authors · 2021-08-09
> **Response to Reviewer#3 2ePa**
>
> Thanks for your nice summary of our work! In the following, we give detailed response to your comments one by one.
>
> >*The method is described well. The experiments show that the method can perform well in terms of quantitative analysis and qualitative comparison. The qualitative comparison shows that the method is well suited for screen content compared to other SOTA methods.*
>
> Reply: Thanks for your positive comments on our writing and experiments!
>
> >*With the help of transformers, the SR seems to improve in quality, but there is no analysis on computational complexity. Normally, when the transformer is used, it can be utilized to reduce the computational load while also improving quality. Does this work also convey such characteristics?*
>
> Reply: Thanks for your suggestion. Table R1 presents the computation cost of our method and compared methods. Regarding inference time, our  model is faster than RCAN and LIIF. Our memory cost is less than that of TTSR. In the future, we would like to optimize the MLP structure to reduce our memory cost and Flops.
>
> Table R1. Comparison of Flops, Inference Time, Memory, and Parameters for $128\times128$ LR input with $\times4$ upsampling.
>
> ||TTSR| EDSR | RCAN |RDN  | LIIF | Our |
> |---|---| --- | --- | --- | --- | --- |
> |GFlops|409 | 824|  261 |  372.7| 723 | 1032 |
> |Inference Time| 0.05s|0.02s | 0.15s |  0.05| 0.12s | 0.09s |
> |Memory| 4723M |1202M| 3755M | 1267M |1471M  |4579M |
> |Parameters| 6.73M|43.09M|15.59 M |22.27M |22.32M | 22.61M|
>
>
> We would like to point out that using transformer generally involves extensive computing resources. [R7] claims that:“*transformers “do not generalize well when trained on insufficient amounts of data” and the training of these models involved extensive computing resources*”; [R8] claims that “*Thus, the increased expressivity of transformers comes with quadratically increasing computational costs, because all pairwise interactions are taken into account. The resulting energy and time requirements of state-of-the-art transformer models thus pose fundamental problems for scaling them to high-resolution images with millions of pixels.”).
>
>
>
> >*The latest SOTA for SR using transformers is the CVPR 2020 paper TTSR which is not mentioned in the main paper. There is no comparison with the method either nor in the related works section. The latest SOTA is essential to show in the paper so that it can be compared to the latest SOTA. Although this work is focused on SCISR, it is still very related to SR and further uses transformers as a component to the network architecture. The most similar work is arguably the TTSR which needs to be addressed. Also, the authors conduct experiments on RDN, RCAN, MetaSR etc. However, these are somewhat outdated SR models. Thus, it is not entirely surprising that the current model outperforms them. The latest SR models that the authors should consider is definitely TTSR, but also EDVR, EDSR, IGNN, SRGAN etc. I would say the experimental section is not as extensive as I would have expected. Although there are not many methods combining transformers with CNNs yet for SR, it is much more relevant to compare with CNN only methods to show the effectiveness for combining an additional module; the transformer.*
>
> Reply: Thanks for mentioning TTSR! We did not compare with TTSR because TTSR is a reference based SR, which requires a similar reference image to assist the SR process of the LR input. The key and value in its transformer module are from the reference image. In contrast, our method is a single image based SR. The key and value in our transformer are from the LR image itself. Nevertheless, we compare with TTSR by utilizing its LR images as the reference images (followed the settings of TTSR for dataset without reference images). As shown in Table R2, we retrain TTSR for $\times4$ upsampling on our SCI1K dataset, and our method greatly outperforms TTSR. The main reason is that TTSR is designed for reference based SR (RefSR) other than single image SR (SISR).
>
>  Table R2. Comparison of $\times4$ SR performance on SCI1K, SIQAD, and SCID datasets. All the methods are trained on the training set of SCI1K.
>
>
> |  | TTSR | RCAN | LIIF | Our |
> | --- | --- | --- | --- | ---|
> |SCI1K| 29.93 | 30.80 | 30.55|30.82|
> | SIQAD |  22.25 |  23.69| 23.44 |24.03 |
> | SCID | 24.91 | 26.02 | 25.89 | 26.22 |
>
> In addition, as shown in Table R3, TTSR is much worse than RCAN on the Urban100 dataset where TTSR is trained with CUFED5 and RCAN is trained with DIV2K dataset (due to limited time for rebuttal, we did not retrain TTSR on DIV2K). In the final version, we will retrain TTSR on DIV2K dataset to further confirm that TTSR is not a good choice for SISR.
>
> Table R3. Comparison of $\times4$​ SR performance for natural images. All the methods (*except TTSR*) are trained on DIV2K dataset and test on Urban100 and B100 datasets.
>
> | Methods |  EDSR[R1] | SRGAN[R2] | RDN | RCAN |SRFBN[R3]  |
> | --- | --- | --- | --- | --- |  --- |
> |  Years| CVPR2017 | CVPR2017 | CVPR2018 | ECCV2018  | CVPR2019  |
> | Urban100 |26.64  | ---- | 26.61 | 26.82 | 26.60 |
> | B100 |27.71  | 25.16 | 27.72 | 27.77 | 27.72 |
>
> | Methods |  TTSR[R4] | IGNN[R5] | NLSN[R6] | MetaSR | LIIF  | Our |
> | --- | --- | --- | --- | --- |  --- | --- |
> | Years| CVPR2020 | NeurIPS2020 | CVPR2021 | CVPR2019 | CVPR2021 ||
> | Urban100 |25.87  |26.84|  26.96 |26.55 |26.68 | 26.73 |
> | B100 |---- |27.77| 27.78 | 27.71| 27.74| 27.72 |
>
>
>
> We did not compare with EDVR since it is designed for video SR while our work is for image SR. In addition, the SR performance of RCAN is better than EDSR, and SRGAN, and is comparable with IGNN on the benchmark natural image dataset, as shown in Table R3. Although RCAN was published in 2018, its performance still outperforms many methods published after 2018. To thoroughly evaluate the proposed method, we compare with 10 benchmark SR methods on natural image SR datasets B100 and Urban100, as shown in Table R3. It can be observed that our method is ranked the fourth on Urban100 dataset and is only 0.06 dB less  than the best method NLSN on the B100 dataset. In addition, our main contribution is the transformer module for single image SR. Our feature extraction backbone can be changed to any SOTA CNN-based SR method to further improve the performance.
>
> >**Limitations And Societal Impact:**
> *SR has the potential to be a piece of technology that can impact the society perhaps in a negative way. SR in the end is trying to construct new pixels or even hallucinate details that were not seen or non existent. This means that the picture may not be entirely true to the true visual. This has a weak link with deep fakes. Some thought on this is required. However, the authors do not address such societal or ethical implications in this paper.*
>
> Reply: Thanks for you suggestion! We will add this limitation and societal impact in our final version.
>
>
> [R1] Bee Lim, Sanghyun Son, Heewon Kim, Seungjun Nah, and Kyoung Mu Lee, "Enhanced Deep Residual Networks for Single Image Super-Resolution," 2nd NTIRE: New Trends in Image Restoration and Enhancement workshop and challenge on image super-resolution in conjunction with CVPR 2017.
>
> [R2] Ledig C, Theis L, Huszár F, et al. Photo-realistic single image super-resolution using a generative adversarial network, CVPR, 2017: 4681-4690.
>
> [R3] Li Z, Yang J, Liu Z, et al. Feedback network for image super-resolution, CVPR, 2019: 3867-3876.
>
> [R4] Yang F, Yang H, Fu J, et al. Learning texture transformer network for image super-resolution, CVPR, 2020: 5791-5800.
>
> [R5] Zhou, Shangchen, et al. “Cross-Scale Internal Graph Neural Network for Image Super-Resolution.” NeurIPS, vol. 33, 2020, pp. 3499–3509.
>
> [R6] Mei, Yiqun, et al. “Image Super-Resolution With Non-Local Sparse Attention.” CVPR, 2021, pp. 3517–3526.
>
> *[R7] Touvron, Hugo, et al. "Training data-efficient image transformers & distillation through attention." International Conference on Machine Learning. PMLR, 2021.*
>
> *[R8] Esser, Patrick, Robin Rombach, and Bjorn Ommer. "Taming transformers for high-resolution image synthesis." CVPR, 2021.*

---

> > ### Comment · Reviewer_2ePa · 2021-08-24
> > **Reply**
> >
> > I have read the other reviews and comments. The authors have addressed my concerns extensively and very well. I am willing to bump up my final score.

---

### Official Review · Reviewer_cjuE · 2021-07-18

**Rating:** 7
**Confidence:** 4

**Summary:**

This paper addresses the problem of arbitrary screen content image super-resolution, which is quite different from natural image super-resolution.  The author proposed two datasets specifically designed for this scenario, which can greatly motivate research  effort into this field. Besides, the authors proposed Implicit Transformer Super-Resolution Network (ITSRN) to achieve arbitrary scaling for screen content images. The ITSRN is largely inspired by LIIF, the main novelty of ITSRN includes the proposed  implicit transformer and the implicit position encoding, these schemes are empirically evaluated to show its effectiveness.

**Main Review:**

Originality: The problem of arbitrary scale super-resulution of screen content image is addressed in this work, which is a specific subarea of image super-resolution and is rarely studied by researchers. The proposed ITSRN is inspired by LIIF and its performance is improved by implicit transformer and implicit positional coding. In my opinion, the task is interesting and the method is novel. The newly proposed datasets can benchmark this subarea and is helpful for the developing of the subarea.

Quality: the quality of this paper is above the bar. Technical details are clear and the evaluation is convincing, extensive results are provided to show the effectiveness of implicit transformer and implicit position encoding, some design choices are also empirically verified. However, the following two issues should be addressed to further improve this paper:
1) what is the complexity of the proposed method, compared to existing state-of-the-arts. This information can help readers know better about the strength of the paper as well as its overhead.
2) as stated in line201-202, "the 9c-dimensional output is mulitplied by the corresponding feature value v, generating the coarse results I_q". It is a little bit confusing how the mulitiplication is conducted, because  v is also a 9c-dimensional vector, i am guessing that I_q is a scalor or not? If yes, how to generate the 3-channel output RGB pixel value at q. Please make this more clear to readers.

Clarity: This paper is well organized and is easy to follow. The writting of this paper is good. Though small ambiguity exists as stated above, codes and models are currently available for verification, which is a plus.

Significance: This work makes a different to the community. Firstly, the datasets can benchmark the subarea; secondly, the problem is interesting; lastly, the method is new and inspiring.

**Time Spent Reviewing:**

48

---

> ### Author Response · Authors · 2021-08-09
> **Response to Reviewer#2 cjuE**
>
> Thanks for your nice summary and positive comments on our novelty, experiments, and datasets! In the following, we give detailed response to your comments one by one.
>
> >*What is the complexity of the proposed method, compared to existing state-of-the-arts. This information can help readers know better about the strength of the paper as well as its overhead.*
>
> Reply: Thanks for your suggestion. Since we utilize MLP, our network consumes more flops and memories. Table R1 lists the computation cost and inference time of our method and compared methods. Regarding the inference time, our model is faster than RCAN and LIIF. Our memory cost is less than that of TTSR. In the future, we would like to optimize the MLP structure to reduce our memory cost.
>
> Table R1 Comparison of Flops, Inference Time, Memory, and #Parameter for $128\times128$ LR input with $\times4$ upsampling.
>
> ||TTSR| EDSR | RCAN |RDN  | LIIF | Our |
> |---|---| --- | --- | --- | --- | --- |
> |GFlops|409 | 824|  261 |  372.7| 723 | 1032 |
> |Infertime| 0.05s|0.02s | 0.15s |  0.05| 0.12s | 0.09s |
> |Memory| 4723M |1202M| 3755M | 1267M |1471M  | 4579M |
> |parameters| 6.73M|43.09M|15.59 M |22.27M |22.32M | 22.61M|
>
>
>
> >*As stated in line201-202, "the 9c-dimensional output is mulitplied by the corresponding feature value v, generating the coarse results I_q". It is a little bit confusing how the mulitiplication is conducted, because v is also a 9c-dimensional vector, i am guessing that I_q is a scalor or not? If yes, how to generate the 3-channel output RGB pixel value at q. Please make this more clear to readers.*
>
> Reply: Sorry for the unclear description. The output size of the MLP is actually 27c, which is reshaped to $9c\times 3$. Then, the feature value $v\in 1\times 9c$ is multiplied with the $9c\times 3$ output, generating a $1\times 3$ value $I_q$, namely the RGB pixel value at q. We will revise the corresponding statements in the final version.

---

> > ### Comment · Reviewer_cjuE · 2021-08-21
> > **Comments on authors' rebuttal**
> >
> > Most of my problems are addressed in the authors' response.
> > I recommend an accept for this paper.

---

### Official Review · Reviewer_bkFN · 2021-07-19

**Rating:** 7
**Confidence:** 3

**Summary:**

The paper proposes a transformer-based image super-resolution method for screen content images which have many thin edges. The transformer is designed to learn the mapping from coordinates to rgb values. It additionally has a scale token representing the magnification factor. Due to targeting a new application, the paper also constructs suitable training and testing datatests.

**Limitations And Societal Impact:**

Yes, the relevant limitations have been discussed.

**Main Review:**

*Originality*

\+ Addresses the problem of super-resolution of screen content images, which have different characteristics than natural images (e.g., many thin edges). Seem to be the first one looking at this specific application.

\+ Allows for continuous super resolution and is not limited to fixed, discrete magnification factors.

\+ Introduces an implicit transformer based super-resolution network.

\+ Construct two kinds of benchmark datasets, once with bicubic downscaling and once adding JPEG compression.

*Quality*

\+ The superiority of the method for the targeted domain of screen images has been shown on its own dataset as well as related datasets constructed for image quality assessment and in comparison, to other SR methods.

\+ Method also works for scale factors which are not used during training.

\+/- Ablation study has been performed to analyze the benefit of the different modules. The scale token, however, seems to be mainly beneficial for in-training scales.

\+/- For completeness, it might be interesting to add the performance of the proposed methods on natural images.

*Clarity*

\+ Code submitted.

\- If available, ground truth could be added to Figure 1.

\+/- It mentions that the graphic memory needs to be sufficient large at test time. What does that mean? What is the inference time?

\- I am wondering how the performance would be if in-training scale are distributed more (e.g., 2,4,8)? Is the method also working for fractional scale factors?

*Significance*

\+ Constructs a benchmark dataset for the specific application domain of super resolution of screen content images.

\+ Shows superior performance on the targeted application domain.

\- Limited to the specific application.


**Time Spent Reviewing:**

4

---

> ### Author Response · Authors · 2021-08-09
> **Response to Reviewer#1 bkFN**
>
> Thanks for your nice summary and positive comments on our novelty and contributions! In the following, we give detailed response to your comments one by one.
>
> >*1. +/- Ablation study has been performed to analyze the benefit of the different modules. The scale token, however, seems to be mainly beneficial for in-training scales.*
>
> Reply: Thanks for your careful review. We agree with you that the gain of scale token is smaller for out-of-training-scale than that for in-training-scale. The main reason is that for larger upsampling scales (e.g. $\times6$​​​ upsampling), the LR input is very smooth and it is hard to predict high-frequency details with the single LR image as input (in our experiments, the LR input size for $\times4$​​​ upsampling is [H/4,W/4] while the LR input size for $\times6$​​​ upsampling is [H/6,W/6], where H and W is the height and width of the ground truth). Therefore, the scale token contributes smaller compared with that for in-training-scale (As shown in Table 3 of the main paper, the SSIM value is improved by 0.0021 and PSNR value is decreased by 0.05 dB for the out-of-training-scale ($\times6$​​​) after introducing the scale token). To verify this, we retrain our model with training scale x1-x2, and test the performance at $\times4$​​​ upsampling. As shown in Table R1, the performance of the model with scale token outperforms the model w/o scale taken by 0.17 dB. This demonstrates that the scale token also benefits the out-of-training-scales.
>
> Table R1. Ablation study for the scale token by using training scale x1-x2.
>
> |                              | with scale token | w/o scale token |
> | ---------------------------- | ---------------- | --------------- |
> | x4(training scale is  x1-x2) | 29.56            | 29.39           |
>
>
> >*2. +/- For completeness, it might be interesting to add the performance of the proposed methods on natural images.*
>
> Reply: Thanks for your suggestion. We retrain our model with the DIV2K dataset and evaluate it on urban100 and B100 datasets. Table R2 presents the results of our method and state-of-the-art natural image SR methods at $\times4$ upsampling. As suggested by Reviewer #3 2ePa, we also present the results of TTSR, which is a reference-based method. Hence, TTSR is trained on the reference-based SR dataset, CUFED5 dataset. The PSNR value is calculated on Y channel in the transformed YCbCr space. It can be observed that, for natural image dataset Urban100, our method still outperforms state-of-the-art continuous SR method LIIF.  On the B100 dataset, our method is only 0.06 dB less than the best method NLSN. It demonstrates that although our method is not the best for natural image SR, but still achieves promising results on natural images. We will present these results on our final version.
>
> Table R2. Comparison of $\times4$​​ SR performance for natural images. All the methods (*except TTSR*) are trained on DIV2K dataset and test on Urban100 and B100 datasets.
>
>
>
>
> | Methods |  EDSR[R1] | SRGAN[R2] | RDN | RCAN |SRFBN[R3]  |
> | --- | --- | --- | --- | --- |  --- |
> |  Years| CVPR2017 | CVPR2017 | CVPR2018 | ECCV2018  | CVPR2019  |
> | Urban100 |26.64  | ---- | 26.61 | 26.82 | 26.60 |
> | B100 |27.71  | 25.16 | 27.72 | 27.77 | 27.72 |
>
> | Methods |  TTSR[R4] | IGNN[R5] | NLSN[R6] | MetaSR | LIIF  | Our |
> | --- | --- | --- | --- | --- |  --- | --- |
> | Years| CVPR2020 | NeurIPS2020 | CVPR2021 | CVPR2019 | CVPR2021 ||
> | Urban100 |25.87  |26.84|  26.96 |26.55 |26.68 | 26.73 |
> | B100 |---- |27.77| 27.78 | 27.71| 27.74| 27.72 |
>
>
>
>
> >*3. If available, ground truth could be added to Figure 1*
>
> Reply: Thanks for your suggestion. The LR input of Figure 1 is generated by $\times4$ downsampling of the ground truth. Therefore, there is only ground truth for the $\times4$ magnification, which will be given in our final version.
>
> >*4.+/- It mentions that the graphic memory needs to be sufficient large at test time. What does that mean? What is the inference time?*
>
> Reply: We would like to point out that current dense prediction networks (such as super-resolution, denoising, and enhancement) usually cost large memory for images with large size. Therefore, if the memory is sufficient, we can directly generate the SR result with the whole LR image as input. Otherwise, we need to split the LR image into patches, and then generate the SR result for each patch sequentially, and finally merge all the patches together to generate a large SR result. In our experiments, for a $512\times512$ image, the memory cost is 4579Mbytes, and the inference time is 0.09s on a 1080TI GPU. Our memory cost is large since our model contains point-wise MLP structures. In the future, we will optimize the MLP structure to reduce the memory cost. The computation cost comparison will be given in our final version, which is also suggested by Reviewer#2 cjuE.
>
> >*5.- I am wondering how the performance would be if in-training scale are distributed more (e.g., 2,4,8)? Is the method also working for fractional scale factors?*
>
> Reply: Thanks for your suggestion. As shown in Table R2, the SR results at $\times6$ and $\times8$ upsampling with training scales {2-8} are better than that with training scale {2-4}. The main reason is that the scales of $\times6$ and $\times8$ are in the range of the second training scales. In addition, the second strategy is also beneficial for $\times10$ upsampling. This reminds us that using wide distributed training scales is better than using narrow distributed training scales. We will add these results in the final version.
>
> Table R3. Comparison of SR performance with different training scales.
>
>
> |training scale x2-x4  |x4 | x6 |x8 |x10|
> | --- | --- | --- | --- | --- |
> | SCI1K  |30.82 | 26.00 | 23.16|21.77|
>
> |training scale x2-x8 | x4  | x6 | x8  | x10  |
> | --- | --- | --- |--- |--- |
> | SCI1K  | 30.83|26.37 |23.59 |  21.86
>
>
>
> As shown in Fig. 1, our method works for fractional scale factors. As shown in Eq. (5), our network has no constraints on the scale token, which can be integer and fractional factors, no matter what the training scale is.
>
>
> >*6. Limited to the specific application.*
>
> Reply: Yes, our method is designed for the specific screen content SR. However, as shown in our response to your second comment, our proposed method is also able to provide promising SR results for natural images.
>
>
>
> [R1] Bee Lim, Sanghyun Son, Heewon Kim, Seungjun Nah, and Kyoung Mu Lee, "Enhanced Deep Residual Networks for Single Image Super-Resolution," 2nd NTIRE: New Trends in Image Restoration and Enhancement workshop and challenge on image super-resolution in conjunction with CVPR 2017.
>
> [R2] Ledig C, Theis L, Huszár F, et al. Photo-realistic single image super-resolution using a generative adversarial network, CVPR, 2017: 4681-4690.
>
> [R3] Li Z, Yang J, Liu Z, et al. Feedback network for image super-resolution, CVPR, 2019: 3867-3876.
>
> [R4] Yang F, Yang H, Fu J, et al. Learning texture transformer network for image super-resolution, CVPR, 2020: 5791-5800.
>
> [R5] Zhou, Shangchen, et al. “Cross-Scale Internal Graph Neural Network for Image Super-Resolution.” NeurIPS, vol. 33, 2020, pp. 3499–3509.
>
> [R6] Mei, Yiqun, et al. “Image Super-Resolution With Non-Local Sparse Attention.” CVPR, 2021, pp. 3517–3526.

---

> > ### Comment · Reviewer_bkFN · 2021-08-25
> > **Comments after rebuttal**
> >
> > Most of the concerns have been addressed satisfactorily in the rebuttal. I appreciate the additional experiments provided in the rebuttal, which support the result of the paper and answered the questions I had. I therefore increased my review rating from originally '6: Marginally above the acceptance threshold' to '7: Good paper, accept'.

---

### Decision · Program_Chairs · 2021-09-27

**Decision:**

Accept (Poster)

**Comment:**

The paper proposes a new model for image super-resolution, which builds on an implicit transformer, to address the resolution enhancement problem in the specific settings of screen-content images. The performance of the model has been extensively studied by the other and improved through the active discussion with the reviewers. The results are quite convincing for the specific SR task under consideration, which has not been commonly studied so far.